# Therapeutic Manuka Honey as an Adjunct to Non-Surgical Periodontal Therapy: A 12-Month Follow-Up, Split-Mouth Pilot Study

**DOI:** 10.3390/ma16031248

**Published:** 2023-02-01

**Authors:** David Opšivač, Larisa Musić, Ana Badovinac, Anđelina Šekelja, Darko Božić

**Affiliations:** 1School of Medicine, University of Pula, Zagrebačka 30, 52100 Pula, Croatia; 2Department of Periodontology, School of Dental Medicine, University of Zagreb, Gunduliceva 5, 10000 Zagreb, Croatia; 3Health Center Zagreb, Runjaninova 4, 10000 Zagreb, Croatia; 4University Dental Clinic, University Hospital Centre Zagreb, Kišpatićeva 12, 10000 Zagreb, Croatia

**Keywords:** periodontitis, manuka honey, nonsurgical periodontal therapy

## Abstract

Periodontitis is recognized as one of the most common diseases worldwide. Non-surgical periodontal treatment (NSPT) is the initial approach in periodontal treatment. Recently, interest has shifted to various adjunctive treatments to which the bacteria cannot develop resistance, including Manuka honey. This study was designed as a split-mouth clinical trial and included 15 participants with stage III periodontitis. The participants were subjected to non-surgical full-mouth therapy, followed by applying Manuka honey to two quadrants. The benefit of adjunctive use of Manuka honey was assessed at the recall appointment after 3, 6, and 12 months, when periodontal probing depth (PPD), split-mouth plaque score (FMPS), split-mouth bleeding score (FMBS), and clinical attachment level (CAL) were reassessed. Statistically significant differences between NSPT + Manuka and NSPT alone were found in PPD improvement for all follow-up time points and CAL improvement after 3 and 6 months. These statistically significant improvements due to the adjunctive use of Manuka amounted to (mm): 0.21, 0.30, and 0.19 for delta CAL and 0.18, 0.28, and 0.16 for delta PPD values measured after 3, 6, and 12 months, respectively. No significant improvements in FMPS and FMBS were observed. This pilot study demonstrated the promising potential of Manuka honey for use as an adjunct therapy to nonsurgical treatment.

## 1. Introduction

Periodontitis is a chronic inflammatory disease affecting the teeth’s supporting apparatus. Bacterial biofilm and the associated periodontal pathogenic bacteria, mainly Gram-negative anaerobes, are the main etiological factor of the disease [1].

The main goal of periodontal treatment is to reduce the number of periodontal pathogens and arrest the inflammatory process. The contemporary gold treatment standard is non-surgical periodontal therapy (NSPT), which involves scaling and root planning using manual and machine-driven (sonic or ultrasonic) instruments [2]. The literature suggests that this therapy is highly effective in eliminating the infection. The latest systematic review article by Suvan et al. on subgingival instrumentation for periodontitis treatment estimates a weighted range of pocket depth reduction of 1.0–1.7 mm and a ratio of pocket closure of 57–74% after 3/4 and 6/8 months, respectively, that was achieved through non-surgical periodontal treatment only [3]. Although NSPT can effectively reduce the number of periodontal pathogens, microbial recolonization commonly occurs, and residual pockets are expected to remain after NSPT [2].

Various systemically administered and locally delivered adjuncts to NSPT have been suggested, including systemic and local antibiotics, antiseptics, probiotics, lasers, and photodynamic treatment. However, the latest guidelines on the treatment of periodontitis stage I–III do not support the use of adjuncts. The exception in terms of open recommendations is given for locally administered sustained-release chlorhexidine and antibiotics and the use of systemic antibiotics in specific patient groups [2].

The fact that bacteria are becoming increasingly resistant to antibiotics and antiseptics has shifted the interest of medicine to alternative treatment methods against which bacterial resistance cannot be developed. This approach includes using honey, which is increasingly used in medicine. Since the 1990s, when the first studies appeared on the therapeutic effects of honey, particular interest has been focused on its antibacterial properties against infections and antibiotic-resistant bacteria. This effect is consequential mainly of the high sugar concentration of honey, its low pH value, and the formation of hydrogen peroxide that occurs in the enzymatic breakdown of glucose by the glucose oxidase enzyme [4]. Contemporary research on the effects of honey focuses predominantly on one specific honey type, leading to the medicinal use of Manuka honey due to its antibacterial properties [5]. This is an endemic type of honey produced by bees in Australia and New Zealand from the flowers of the plant *Leptospermum scoparium* [6].

The concentration of hydrogen peroxide in Manuka honey is lower than in other types of honey [7]. The specific antibacterial activity in Manuka honey is based on methylglyoxal (MGO), a compound proven to be a very efficient bactericide, virucide, and fungicide. Furthermore, Manuka honey is highly effective against antibiotic-resistant bacteria [8]. The antibacterial potency of Manuka honey was found to be related to its Non-Peroxide Activity (NPA), trademarked as Unique Manuka Factor (UMF) rating, a classification system which reflects the equivalent concentration of phenol (%, *w*/*v*) required to produce the same antibacterial activity as honey, and it is correlated with the methylglyoxal and total phenols content [9]. In addition to its antimicrobial properties, published literature suggests that MGO also has immunomodulatory effects which may positively impact wound healing and tissue regeneration [10,11].

Therapeutic Manuka honey has not yet been investigated as a possible adjunct to NSPT. Therefore, this pilot study aims to evaluate the effects of a product containing Manuka honey on periodontal parameters when applied to periodontal pockets after nonsurgical periodontal treatment in patients with stage 3 periodontitis. 

## 2. Materials and Methods

### 2.1. Experimental Design

This study was designed as a single-center prospective pilot trial with a 12-month follow-up. A split-mouth study model was used. Two quadrants were randomly assigned to the test treatment of NSPT + product containing Manuka honey or NSPT-only.

This pilot study was approved by the Ethics Committee of the School of Dental Medicine, University of Zagreb, Croatia; approval No. 05-PA-30-IX-9/2019. All parts of the study were conducted in full accordance with the World Medical Association Declaration of Helsinki on ethical principles for medical research involving human subjects.

### 2.2. Population Screening and Inclusion

Patients who sought or were referred for periodontal therapy at the Clinical Department of Periodontology, University Hospital Zagreb, between September 2019 and March 2021, were screened for possible inclusion in the study. The inclusion criteria were: (1) systemically healthy patients of both genders, between the age of 18 and 70; (2) non-smokers; (3) presence of at least 20 teeth; and (4) untreated generalized advanced chronic periodontitis according to the 1999 Classification 1999 [12], i.e., generalized stage III periodontitis according to the 2007 Classification [13]. Exclusion criteria were: (1) pregnant and nursing women; (2) antibiotics prescribed for dental or non-dental diseases six months before the start of the research; (3) systemic diseases or the use of drugs known to affect periodontal tissues; and (4) acute oral or periodontal inflammation or infection (pericoronitis, necrotizing periodontal diseases, etc.). Following inclusion, a periodontal examination was performed by one calibrated periodontist (D.B.). Assessments were done at six sites using a UNC-15 periodontal probe (HuFriedy, Chicago, IL, USA). The following parameters were measured and recorded: probing pocket depth (PPD), recession of the gingival margin (REC), clinical attachment loss (CAL; calculated as the sum of PPD and REC), split-mouth bleeding score (SMBS; calculated as the percentage of positive bleeding sites on probing and expressed for NSPT + Manuka and NSPT-only quadrants, respectively) and split-mouth plaque score (SMPS; calculated as the percentage of sites with present plaque and expressed for NSPT + Manuka and NSPT-only quadrants, respectively) [14,15]. Third molars, if present, were excluded from data analysis. 

All participants have given written informed consent to study participation.

### 2.3. Periodontal Treatment

Nonsurgical treatment was performed by standardized protocol by a single operator (D.O.). All patients received identical oral hygiene instructions, presuming the use of appropriately sized interdental brushes and manual toothbrushes with regular fluoride-containing toothpaste. The use of mouthwashes of any formulation was not allowed during the study period. Mechanical subgingival instrumentation was performed using an ultrasonic instrument (Piezon, E.M.S. Electro Medical Systems S.A., Nyon, Switzerland) and curettes (BioGent, Hu-Friedy, Chicago, IL, USA). Local anesthesia (Ubistesin 40 mg/mL + 0.005 mg/L, 3M Deutschland GmbH, Seefeld, Germany) was provided to all participants. Instrumentation was performed according to the individual situation and without any time limitation. All treatments were concluded within the timeframe of 24 h.

The adjunctive treatment used in this study was a novel commercial product (Pocket Protect, CleverCool B.V., Lijnden, The Netherlands) containing therapeutic Manuka honey and hydrogen peroxide. The two substances mix within a double-barrel syringe before deposition within the pocket.

The product was administered in all pockets (depths ≥ 4 mm) in the two active quadrants as per the instructions of the manufacturer. Once the inserted syringe reached the bottom of the periodontal pocket, the product was extruded until the excess was observed in the sulcus. Subjects were not allowed to consume drinks or food for at least 30 min following the procedure.

All subjects were required to report possible adverse effects.

The patients were scheduled for recall visits after three, six, and 12 months when PPD and CAL were re-evaluated. In the first three months only were the patients scheduled for supportive treatment at one-month intervals. The supportive treatment consisted of OH re-instructions, if deemed necessary, and supragingival scaling and polishing. The collected data were pseudo-anonymized immediately after collection. Only the clinician performing the treatment had the access to the patient’s identifying information. 

### 2.4. Randomization and Blinding

Randomization of patients was done using a computerized random number generator. Each quadrant was allocated to receive one of the two treatments (Manuka + NSPT or NSPT-only), with the allocation ratio forced to 1:1. The concealment was achieved using sealed and numbered envelopes. A researcher not involved in the operative phases of the study performed the random allocation sequence and intervention assignment.

Blinding was not possible during the experimental period (operator, subjects) due to the specific design of the product-delivery syringe and product’s taste.

The examiner was unaware of the treatment allocation at any point during the ongoing study period. Blinding was also done for statistical analysis.

### 2.5. Statistical Analysis

As the assumption of normality of distribution was verified by inspecting normal Q-Q plots, the comparisons between the NSPT + Manuka and the NSPT-only quadrants were performed using a two-tailed t-test for independent samples with the assumption of homoskedasticity. PPD and CAL values were compared between the NSPT + Manuka and the NSPT-only quadrants at each time point (baseline, three months, six months, and 12 months). The changes in the parameters PPD and CAL (denoted as delta PPD and delta CAL, respectively) were calculated for each time point by subtracting the baseline values from the values measured after three months, six months, and 12 months. The obtained delta values were statistically compared between the NSPT + Manuka and the NSPT-only quadrants using a two-tailed t-test for independent samples. BoP index and plaque index were represented as percentages of sites that were positive for bleeding or the presence of plaque, respectively. These percentages of positive sites were compared between the NSPT + Manuka and the NSPT-only quadrants using the chi-square test.

The statistical analysis was performed using SPSS (version 25; IBM, Armonk, NY, USA) at a level of significance of 0.05.

## 3. Results

A total of 86 patients were screened for inclusion and 15 participants (eight males and seven female) were recruited for this pilot study. However, three of them were excluded from the study due to not showing up at the follow-up appointment at 3 months (two participants) and 6 months (one participant). Hence, a total of 12 participants (five males and seven female) were included. The mean age was 43.1 years (range 31–49), with a mean number of teeth at 26, and 1331 sites with increased PPD (≥4 mm); 4–6 mm at 905 sites and >6 mm at 426 sites. All patients were non-smokers.

Table 1 shows the PPD and CAL values, and the corresponding delta values representing differences from baseline measured after 3, 6, and 12 months. There were no statistically significant differences in baseline values for PPD and CAL between the sites treated with NSPT + Manuka and the sites treated with NSPT-only. In addition, no significant differences were observed for PPD and CAL in the comparisons with the sites treated with NSPT + Manuka and the sites treated with NSPT-only at each of the follow-up time periods (3, 6, and 12 months). However, statistically significant differences between NSPT + Manuka and NSPT-only were identified in delta PPD values for all follow-up time points, as well as for delta CAL values for the time points of 3 and 6 months. The delta CAL values calculated for 12 months could be considered marginally significant at the selected level of significance of 0.05. These statistically significant further improvements due to the adjunctive use of Manuka amounted to (mm): 0.21, 0.30, and 0.19 for delta CAL and 0.18, 0.28, and 0.16 for delta PPD values measured after 3, 6, and 12 months, compared to NSPT-only.

Table 2 shows FMPS and FMBS values measured at baseline, 3, 6, and 12 months. At baseline, the quadrants that received NSPT-only had significantly greater values of plaque and bleeding. No significant differences between the quadrant groups were identified, with the exception of lower bleeding scores at 6 months in the NSPT + Manuka quadrants.

One patient reported generalized dentine hypersensitivity that spontaneously decreased until the 1-month check-up. No other adverse effects were reported.

## 4. Discussion

This pilot study investigated the impact of Manuka honey as an adjunct to NSPT. It showed statistically significant improvements in terms of PPD reduction and CAL gain after 3, 6, and 12 months of follow-up, compared to the outcomes of the NSPT-only. Improvements in bleeding and plaque scores were also observed in both quadrant groups.

The present study was motivated by numerous literature reports on the beneficial effects of Manuka honey when used for various medical purposes. Accelerated wound healing was reported after the topical use of Manuka honey for the treatment of ulcers, bed sores, and other skin infections [4]. In addition, Manuka honey was demonstrated to promote healing in infected wounds that do not respond to conventional therapy, i.e., antibiotics and antiseptics, including wounds infected with methicillin-resistant *S. aureus* [16]. Manuka honey may promote the repair of the damaged intestinal mucosa, stimulate the growth of new tissues, and work as an anti-inflammatory agent [16]. Clinical observations have reported reduced symptoms of inflammation when Manuka honey is applied to wounds [17]. The removal of exudate in wounds dressed with honey was helpful for managing inflamed wounds [4]. The aforementioned effects encouraged our investigation of the possible effects of Manuka honey on periodontitis treatment. 

Although the effects of Manuka honey on oral bacteria have not been investigated in vivo, there are several in vitro studies that convincingly indicated its antibacterial activity. The study by Safii et al. was based on the minimum amount of honey required to kill bacteria or inhibit their growth on blood agar, as evaluated by the minimum bactericidal concentration or minimum inhibitory concentration (MBC/MIC), and showed a high antibacterial potential of Manuka honey, especially against Gram-negative anaerobic bacteria [18]. Similar results were published in the in vitro work of Schmidlin et al. on the antimicrobial effect of Manuka honey compared to other types of honey against three common pathogens present in the oral cavity (*S. mutans*, *P. gingivalis*, and *A. actinomycetemcommitans*). Antibacterial activity was analyzed on blood agar, and Manuka honey demonstrated a stronger antibacterial effect compared to other types of honey due to its specific non-peroxidase activity mediated by methylglyoxal, to which *P. gingivalis* was found to be especially sensitive [17].

The research by Badet and Quero analyzed in vitro adhesion of oral cavity bacteria exposed to different concentrations of Manuka honey on glass and hydroxyapatite surfaces immersed in saliva. The results showed that Manuka honey with higher methylglyoxal concentrations inhibits the adhesion of *S. mutans* [19]. Sigrun et al. compared the effectiveness of Manuka honey and regular honey on *P. gingivalis*, which is one of the most important periodontal pathogenic bacteria. It was shown that the planktonic form is extremely sensitive to exposure to Manuka honey, which additionally substantiates its antibacterial properties [20].

The only in vivo study on the use of Manuka honey in the oral cavity is reported in the article by Al-Khanati et al. who investigated the analgesic effect of Manuka honey applied to the post-extraction alveolus following the extraction of impacted third molars. Their randomized split-mouth study showed a statistically significant reduction in postoperative pain on the side where Manuka honey was applied [21]. This finding suggests favorable analgesic properties and potential anti-inflammatory activity of Manuka honey.

A recent systematic review and meta-analysis analyzed the adjunctive effects of locally delivered antimicrobials in the nonsurgical treatment of periodontitis. The meta-analysis for studies of 6–9-month follow-ups showed statistically significant benefits in terms of PPD reduction (WMD = 0.365) and CAL gain (WMD = 0.263). For long-term studies of 12 to 60 months, significant differences were only observed for PPD (WMD = 0.190). The authors highlighted that the heterogeneity was significant due to a great number of different active agents and a difference in study design. The products with the largest observed benefits were those containing doxycycline or tetracycline [22]. The improvements in the treatment outcome following the adjunctive use of Manuka honey in the present study can be compared to the systematic review’s results in terms of PPD reduction, as the mean difference at 6 and 12 months were 0.30 and 0.19, respectively.

Although the differences between the two quadrant groups in our study seem small and that the baseline values might also seem low, it is interesting to note that in a recent multicenter clinical trial investigating the flapless application of enamel matrix derivative in non-surgical therapy when the authors sub-analyzed CAL changes at 12 months including all pockets, and not only the deep ones, the difference between the control and test group was only 0.1 mm in favor of the enamel matrix derivative, with overall CAL changes of 0.8 mm and 0.9 mm, respectively. However, when they analyzed only pockets of 5–8 mm, CAL changes were significantly higher, 2.1 mm for the control group and 2.2 mm for the test group [23]. If we compare the results from the above-mentioned study on all sites to ours, we can see that our CAL changes in the test quadrants were 1.64 mm and 1.48 mm in the control quadrants which is higher than in the Schallhorn study.

Another observation in our study was the continuous improvements in both PPD reductions and CAL values at 3, 6, and 12 months. However, statistical significance between the two groups was only found at 6 months in favor of the test quadrants for both variables. At 3 months, PPD changes were −1.46 mm for the test quadrants and −1.25 mm coming very close to statistical significance (p = 0.002) with a further improvement to −1.74 mm in test quadrants and −1.55 mm in the control quadrants at 12 months with p values still not significant. This difference in the lack of significance at 3 and 12 months compared to 6 months, although the actual numbers in differences are 0.21 mm, 0.30 mm, and 0.19 mm, could be due to the pressure during probing which cannot be entirely controlled although clinical measurements were done by an experienced and calibrated periodontist.

Both quadrant groups presented with an initial reduction in plaque and bleeding scores following treatment. After 3 months, at re-evaluation, the reduction of bleeding was 35.4% in the Manuka + NSPT and 42.8% in NSPT-only quadrants. These values are lower than previously reported by the systematic review of Suvan et al., of a weighted mean reduction of 56.7% at 3/4 months after non-surgical treatment [3]. At 6 months, while the Manuka + NSPT quadrants presented a further, albeit minimal, reduction of 1.9% in bleeding scores, the NSPT-only quadrants presented a rise in bleeding scores of 7.1%. From 6 to 12 months, quadrants treated with both treatment modalities presented with an increase in bleeding scores. The observed reduction of plaque (SMPS) at 3 months was 61.8% in the Manuka + NSPT quadrants and 64.2% in the NSPT-only quadrants, which is as expected following nonsurgical treatment [24]. At 6 and 12 months, quadrants treated with both modalities expressed an increase in plaque score. The increase in both parameters at later follow-ups could be explained by the omission of intensive supportive treatment at 1-month intervals and lack of patients’ motivation. It needs to be highlighted that the data on the effects of these two treatment modalities and the direct comparison between the two treatments on plaque and bleeding have to be interpreted with caution, as there was a statistically significant difference in these parameters between the Manuka + NSPT and NSPT-only quadrants at baseline. As per the effects of Manuka honey on plaque and bleeding, a pilot study by Molan et al. showed that Manuka honey has a positive effect on reducing the amount of dental plaque and the incidence of gingivitis compared to the control group that did not use Manuka honey [7].

The commercial product used in this research combines Manuka honey and hydrogen peroxide. Thus, the effect of the adjunctive cannot be attributed solely to Manuka. The extent of these substances’ sole or combined impact on clinical outcomes would need to be evaluated in an appropriately designed RCT. Schmidlin et al., however, reported a greater antibacterial effect of Manuka above the NPA value of 15 compared to “regular” honey, whose activity relies on peroxide-based antimicrobial factors [17]. Hydrogen peroxide as an adjunct to non-surgical treatment for pocket irrigation failed to show further improvements in clinical outcomes when compared to subgingival debridement alone [24]. Research on the use of 1.7% hydrogen peroxide in custom trays and H_2_O_2_ photolysis (irradiation of 3% H_2_O_2_ with 405 nm light or diode laser) showed promising results in terms of significantly greater PPD reduction in test groups compared to controls [25,26,27,28].

Adding hydrogen peroxide in the commercial product definitely facilitates clinical handling as it decreases the honey’s viscosity and the application inside the periodontal pockets.

The main limitation of this pilot study is the small sample size (n = 12). However, despite the small sample size, the included subjects represent a specific population of patients with severe inflammation and advanced forms of periodontitis, for which it was possible to identify a small but statistically significant improvement in the reduction of PPD at sites treated with Manuka honey compared to the control sites. An analogous improvement was identified for CAL in the sites treated with Manuka honey.

Furthermore, microbiological analysis was not conducted in this research. So far, antibacterial efficacy of Manuka on periodontal pathogens was shown only in in vitro studies [17,18]. Thus, further data is needed from studies on humans. Studies with mid- and long-term follow-ups may show possible prolonged effects of this adjunctive on the subgingival microbial composition compared to subgingival debridement alone.

The statistical analysis was performed at the level of individual periodontal sites, meaning that the statistical unit was represented by the site instead of the patient. When individual sites are considered as statistical units, the actual sample size becomes 2025 (12 subjects × 28.125 teeth per subject on average × 6 sites per tooth = 2025 sites in total). Calculating the “delta” values for PPD and CAL and using these values instead of the original pooled PPD and CAL values measured at each individual time point provided higher statistical power for identifying significant differences in the comparisons of the NSPT + Manuka and the NSPT-only quadrants. By calculating the differences in PPD and CAL at the level of the individual site, the heterogeneity among sites becomes less influential, as the “net” effect is isolated for each site and used for the statistical analysis rather than pooling the original PPD and CAL values measured at individual time points. This reasoning explains why the significant effects of Manuka honey were identified only for delta PPD and delta CAL variables and not for the original PPD and CAL variables. Furthermore, the authors are aware of the possible disadvantages of the split-mouth design. As both sides received some modality of treatment and the product itself comes in a highly viscous form and is delivered locally, the possibility of the carry-across effect (i.e., bacterial contamination from an untreated sides/possible effect of the Manuka honey on the NSPT-only quadrants) was minimized [29,30]; however, we do recognize a risk of it happening. A relatively high number of teeth (20 teeth) set as an inclusion criterion also positively influenced the similarity between randomization units.

## 5. Conclusions

This pilot study indicated a promising potential of Manuka honey as an adjunct in NSPT. Despite the improvement in outcomes appearing modest in absolute terms, it was statistically significant for all follow-up time points, indicating the potential use of Manuka honey as a simple and affordable adjunct to non-surgical periodontal therapy. Furthermore, the product is considered a safe adjunct and no adverse events related to its use were reported during the study period. Encouraging results from this pilot study led to a further randomized clinical study on a larger sample that is currently being performed by our group.

## Figures and Tables

**Table 1 materials-16-01248-t001:** Measured values of periodontal pocket depth, clinical attachment level, and changes of these parameters from the baseline values. All data are represented as mean values with 95% confidence interval limits in parentheses.

Variable	Time Point	NSPT + MANUKA	NSPT-Only	*p*-Values
PPD (mm)	baseline	4.27 (4.12, 4.43)	4.07 (3.93, 4.21)	0.060
	3 months	2.81 (2.72, 2.91)	2.82 (2.73, 2.92)	0.846
	6 months	2.57 (2.48, 2.65)	2.66 (2.57, 2.74)	0.140
	12 months	2.53 (2.45, 2.61)	2.52 (2.44, 2.60)	0.825
∆PPD (mm)	3 months	−1.46 (−1.56, −1.36)	−1.25 (−1.34, −1.15)	**0.002**
	6 months	−1.71 (−1.81, −1.60)	−1.41 (−1.51, −1.31)	**<0.001**
	12 months	−1.74 (−1.85, −1.63)	−1.55 (−1.66, −1.44)	**0.016**

CAL (mm)	baseline	4.30 (4.14, 4.45)	4.11 (3.97, 4.25)	0.080
	3 months	2.91 (2.81, 3.02)	2.91 (2.81, 3.01)	0.952
	6 months	2.69 (2.60, 2.79)	2.78 (2.68, 2.88)	0.201
	12 months	2.66 (2.57, 2.75)	2.63 (2.54, 2.72)	0.609
∆CAL (mm)	3 months	−1.38 (−1.49, −1.28)	−1.20 (−1.30, −1.10)	**0.012**
	6 months	−1.61 (−1.72, −1.49)	−1.33 (−1.43, −1.22)	**<0.001**
	12 months	−1.64 (−1.72, −1.52)	−1.48 (−1.59, −1.37)	**0.052**

PPD—periodontal pocket depth; CAL—clinical attachment level; ∆—delta.

**Table 2 materials-16-01248-t002:** Measured values of split-mouth bleeding score and split-mouth plaque score, represented as percentages of positive sites in respective quadrants (%).

Variable	Time Point	NSPT + MANUKA	NSPT-Only	*p*-Values
SMBS (%)	baseline	82.1	86.5	**0.007**
	3 months	46.7	43.7	0.178
	6 months	44.8	50.8	**0.007**
	12 months	54.6	55.8	0.575
SMPS (%)	baseline	92.7	95.4	0.009
	3 months	30.9	31.2	**0.990**
	6 months	46.7	47.4	0.775
	12 months	51.3	52.2	0.678

SMBS—split-mouth bleeding score (number of positive sites/all measuring sites × 100 in Manuka or NSPT-only quadrants); SMPS—split-mouth plaque score (number of positive sites/all measuring sites × 100 in Manuka or NSPT-only quadrants); %—percentage.

## Data Availability

Not applicable.

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
