# Peer review of "Therapeutic Manuka Honey as an Adjunct to Non-Surgical Periodontal Therapy: A 12-Month Follow-Up, Split-Mouth Pilot Study"

_materials, 2023, doi:10.3390/ma16031248_

Round 1
Reviewer 1 Report
I appreciate the opportunity to evaluate the article titled "Therapeutic Manuka honey as an adjunct to non-surgical periodontal therapy: a 12-month follow-up, split-mouth pilot study". It is a report on a pilot study on the use of a new Manuka honey product for treatment of periodontitis. It is well-written and covers all of the important aspects of a pilot study. Nevertheless, I noticed a few minor issues in the manuscript, which I note for possible improvement. The authors may, at their discretion, take them into account and, specifically, underline in the discussion the requirements necessary for the implementation of the full RCT, not simply a pilot trial.
INTRODUCTION
Besides antimicrobial effects, please consider introducing also Manuka honey's wound healing capacity.
MATERIALS AND METHODS:
- Inclusion criteria: please add the target age range of the subjects.
- Screened patients: how many patients were screened altogether?
- Please consider including Grade of periodontitis to the abstract and methods.
- Line 88: please reword “Patients of both genders with good systemic health” to “Systemically healthy patients of both genders”.
- Line 98: please change “pocket probing depth (PPD)" to “probing pocket depth (PPD)”.
- Line 102: please change “for” to “from”.
- Line 107: please change “DO” to “D.O.”. Please specify what oral hygiene instructions were given.
- Provide more details regarding randomization, allocation concealment and blinding. Please add details regarding the randomisation of the four mouth quadrants in your split- mouth design.
- Line 112: please specify the type of local anaesthesia used.
- Line 127: please specify the type and mode of supportive treatment.
RESULTS:
- Please add more data regarding the patient characteristics at baseline (% of smokers, x̄ no. of teeth, x̄ no. of sites with increased PD …). Also include a measure of variability.
- Line 167: please use the same tense in the sentence “the quadrants that received NSPT only have significantly greater values”. Please revise the whole paper for similar errors.
- Table 2: percentage symbols are not necessary in the table as you defined the unit in the title of the table. Instead, add the percentage symbol next to FMBS and FMPS: FMBS (%), FMPS (%).
- Was a greater effect/improvement observed at sites with moderate or deep initial PPDs?
- Split-mouth design requires appropriate FMBS and FMPS modifications (split mouth BS/PS instead of full mouth BS/PS?).
- Please add data on the reported adverse effects.
DISCUSSION:
- Lines 194-206: In vitro studies only evaluated Manuka honey activity against single-species bacteria, not complex oral biofilms. Please discuss.
- Please discuss the lack of microbiologic sampling as a limitation of the study.
- Lines 219 – 232: please shorten this paragraph. Instead of comparing Manuka honey to systemic antibiotics, it would be more relevant to compare it to locally administered antibiotics/antimicrobials.
- Line 237: when comparing chlorhexidine to Manuka honey, the paper states that chlorhexidine “values are of considerable practical relevance.” Please discuss the clinical significance of a 0.2 mm statistically significant difference in PPD.
- Please discuss the formulation of Pocket Protect (Manuka honey + hydrogen peroxide + …). The effect on the outcome of this study can be partially attributed to the hydrogen peroxide in the commercial product. Please compare your results to clinical studies using pure hydrogen peroxide as a pocket irrigant after non-surgical periodontal therapy.
- No placebo was used. Please discuss the limitations.
- Site-level statistical analysis has limitations due to nesting of measuring sites to teeth and teeth to a patient. Please discuss also other possibilities of statistical analysis (e.g. multilevel statistical model described by Mombelli et al.), that is, however, not possible with a current pilot study design.
Reviewer 2 Report
The paper presented a novel adjunctive therapy to the non surgical therapy to periodontitis stage III patients. The authors presented a pilot study with few patients trying to evaluated the positive impact of the Manuka honey in the non surgical treatment. The Methods and Materials are adequate to the purpose, but with small amount of patients.
The results are well and clearly presented with text and tables which makes the reader easy to understand. In the results they emphasized the possible advantages by using Manuka honey and adjunctive to periodontal non surgical treatment. When compared the control and test group during the follow-up period, the results shows a a slight better results in the test group, with ES in the 3 and 6 months follow-up. Anyhow this ES was not presented in the 12 months follow up. With respect to this, I would like to see a possible explanation for that in the discussion.
In fact, when we look to the table number 1, the PPD in test group was reduced from 4.27 to 2.53 and in the controle group from 4.07 to 2.52, at 12 months of follow-up. In fact, the reduction to achieve normal results was quite similar in both groups.... Another important topic that I think would be good to discuss a little more in the discussion was the baseline values. The mean of the values are quite low for both groups... could the differences be higher if the baseline values were high?
To conclude I think this is a pilot study well design that evaluate a new possibility to improve the results of the non-surgical therapy in stage III Periodontitis patients, with a novel product, that seems to don’t have adverse effect and contra-indications.
